# Intriguing Properties of Deep Neural Policy Manifold: Intrinsic Correlation and Deep Neural Policy Curvature

## Abstract

The progress in deep reinforcement learning research has allowed us to construct policies that can solve complex problems in high-dimensional MDPs leading to the creation of AI agents that can strategize and reason solely by interacting with an environment without any supervision. Yet, the knowledge we have on the underlying structure of the deep neural policy manifold is limited. In this paper, we discover that there is a strong correlation between the advantage function and the gradient of the loss targeting directions of instability. By leveraging this intrinsic correlation, we propose a novel algorithm that can diagnose deep neural policy decision volatilities when their environment contains instabilities. We provide theoretical foundations for this intrinsic correlation, and we conduct extensive empirical analysis in the Arcade Learning Environment with high-dimensional observations. From algorithmic and architectural changes to natural distributional shifts and worst-case perturbations, our proposed method can identify and diagnose the differences by leveraging the intrinsic correlation. Our analysis reveals foundational properties of the deep neural policies trained in high-dimensional MDPs, and our work, while laying the groundwork for reliability, is further a fundamental step towards constructing stable and generalizable policies.

## 1 Introduction

From algorithmic to architectural advances, deep reinforcement learning research has led to building policies that can solve tasks purely from high-dimensional observations (Mnih et al., 2015). From highly complicated board games (Schrittwieser et al., 2020) to foundation models, and with diverse risk-critical settings from finance to pharmaceuticals, deep reinforcement learning is widely installed in many different fields (Fawzi et al., 2022; Mankowitz et al., 2023; Krishnamurthy et al., 2024; Su et al., 2025). On the other hand, while the capabilities of policies have experienced accelerated growth over the years, the knowledge we have on the functions learnt by deep reinforcement learning policies and the structure of the deep neural policy loss landscape is strictly limited.

In this direction, a line of research focused on demonstrating volatilities and instabilities of the models via introducing perceptually indistinguishable perturbations, i.e. adversarial, to the input (Szegedy et al., 2014; Goodfellow et al., 2015; Madry et al., 2017). Not only do these directions of instability raise serious concerns on the robustness and stability of deep neural policies, further the training techniques that are proposed to overcome these problems result in base performance loss (Gourdeau et al., 2019), generalization issues (Korkmaz, 2023), invariance to ground truth semantically meaningful changes (Tramèr et al., 2020; Kumano et al., 2024) and computational intractability (Bhagoji et al., 2019), while these directions further reveal surprising geometric properties of the policy loss landscape (Korkmaz & Brown-Cohen, 2023). Thus, it is of crucial importance to understand the underlying geometry of the deep neural policy manifold, both to develop scientific understanding of the functions learnt by these policies, and to diagnose and mitigate the risks already identified. In this paper, we study geometric properties of the directions of instability in the deep reinforcement learning loss landscape, and how they relate to the advantage function, a fundamental quantity computed and utilized widely across reinforcement learning algorithms. Investigating this geometry reveals an intriguing relationship which we develop further to make the following contributions.

**Contributions.** We discover that there is an intrinsic correlation between the negative average of the advantage function and directions of instability in deep reinforcement learning. We provide foundational analysis to investigate and characterize the intrinsic correlation. We first theoretically unravel the discovered phenomenon by providing a mathematically rigorous relationship between approximate convexity of the negative average advantage and the observed correlation in Section 3. We propose an algorithm to identify the uncertainties within an MDP by leveraging the intrinsic correlation. We then conduct extensive experiments in the Arcade Learning Environment in Section 4, and our results verify the theoretical analysis and demonstrate that we can diagnose the environment the policy is in via leveraging this intrinsic correlation. Our work shows that from natural non-robust directions to adversarial directions the intrinsic correlation gives away the signal of the presence of instabilities. We further investigate different algorithms and architectural changes and the intrinsic correlation between negative average advantage and directions of instability. Our analysis further demonstrates that algorithms that perform better have higher correlation between these values. Thence, our results reveal a profound and pivotal property of the deep reinforcement learning manifold that there is a strict connection between performance of a deep neural policy and the approximate convexity of the negative average advantage.

## 2 BACKGROUND AND PRELIMINARIES

**Preliminaries.** A Markov Decision Process is represented as a tuple $\mathcal{M} = \langle \mathcal{S}, \mathcal{A}, \mathcal{P}, r, \gamma, \mu_0 \rangle$ where $\mathcal{S}$ represents the state space, $\mathcal{A}$ represents the actions space, $r(s, a) : \mathcal{S} \times \mathcal{A} \rightarrow \mathbb{R}$ is the reward function in which provides the rewards received when in state $s$ the policy takes action $a$ and transitions to state $\hat{s}$, $\gamma$ represents the discount factor to prioritize short-term vs long-term rewards and $\mathcal{P}(s, a, \hat{s})$ on $\mathcal{S} \times \mathcal{A} \times \mathcal{S}$ is the transition probability function. The objective in reinforcement learning is to learn an optimal policy via interacting with an environment $\varepsilon$ to maximize the expected cumulative rewards $\mathcal{R} = \mathbb{E}_{a_t \sim \pi(s_t, \cdot), s_{t+1} \sim \mathcal{P}(s, a, \cdot)} \sum_t \gamma^t r(s_t, a_t)$ obtained by the policy $\pi(s, a)$. $\mathcal{Q}$-learning achieves this objective via iterative Bellman update (Bellman, 1957). Precisely, $\mathcal{Q}$-learning is

$$\mathcal{B}^\pi \mathcal{Q}(s, a) = \mathbb{E}[r(s, a)] + \gamma \mathbb{E}_{a \sim \pi(s, \cdot), \hat{s} \sim \mathcal{P}(s, a, \cdot)} \max_{\hat{a}} \mathcal{Q}(\hat{s}, \hat{a}).$$

$\mathcal{B}^\pi$ represents the *Bellman operator* and $\mathcal{V}(s) = \max_a \mathcal{Q}(s, a)$ determines the value of the state.

**Computation of Adversarial Directions.** The vulnerabilities of deep neural networks have been demonstrated in the early work of Szegedy et al. (2014). By adding imperceptible perturbations computed by a box-constrained optimization to the sample, the authors were able to change the decision of the model. While this method is computationally draining, later the work of Goodfellow et al. (2015) proposed the fast gradient sign method via linearizing the cost function at the sample where the adversarial direction that needs to be computed. Further, this objective has been extended to an iterative version by Kurakin et al. (2016).

$$x_{\text{adv}}^{K+1} = \text{clip}_\epsilon(x_{\text{adv}}^K + \alpha \text{sign}(\nabla_x J(x_{\text{adv}}^K, y)))$$

Another natural extension of optimizing this objective was first extended to momentum iterative Dong et al. (2018), and later is computed via Nesterov Momentum proposed in Ezgi (2020) in deep reinforcement learning

$$v_{t+1} = \mu \cdot v_t + \frac{\nabla_{s_{\text{adv}}} J(s_{\text{adv}}^t + \mu \cdot v_t, a)}{\|\nabla_{s_{\text{adv}}} J(s_{\text{adv}}^t + \mu \cdot v_t, a)\|_1} \quad \text{and} \quad s_{\text{adv}}^{t+1} = s_{\text{adv}}^t + \alpha \cdot \frac{v_{t+1}}{\|v_{t+1}\|_2}$$

where $\mu$ is the decaying factor. Another line of adversarial direction computation is focused on computing the minimum distance to a decision boundary where the optimal decision of the policy is not executed under the adversarial direction. In particular, Carlini & Wagner (2017) formulation targets to optimize the objective

$$\min_{x \in \mathcal{X}} c \cdot J(x_{\text{adv}}) + \|x - x_{\text{adv}}\|_2^2$$

The elastic-net regularization (i.e. ENR) is the same objective of the Carlini & Wagner (2017) formulation with the $\ell_1$-norm regularized version (Chen et al., 2018).

$$\min_{x \in \mathcal{X}} c \cdot J(x_{\text{adv}}) + \kappa_1 \|x - x_{\text{adv}}\|_1 + \kappa_2 \|x - x_{\text{adv}}\|_2^2$$

**Issues with Robustness and Functional Instabilities of Deep Reinforcement Learning.** Early studies in the adversarial view demonstrated that deep neural policies are vulnerable to imperceptible perturbations introduced to their input (Huang et al., 2017) computed via the fast gradient sign method Goodfellow et al. (2015). Following this line of work several studies proposed further optimization techniques to compute these adversarial directions (e.g. Nesterov Momentum). Further studies demonstrated via the Carlini & Wagner (2017) and ENR formulation (Chen et al., 2018) that deep reinforcement learning policies learn adversarial features that can be shared across MDPs and across algorithms including certified robust trained policies (Ezgi, 2022). To address these volatility issues of deep neural policies a large body of work focused on techniques to obtain robustness. In particular, Pinto et al. (2017); Gleave et al. (2020) proposed to model this interaction as zero-sum Markov game and jointly train the policy in the presence of an adversary. Yet, recent work demonstrated certain geometric properties of the deep neural policy landscape Korkmaz & Brown-Cohen (2023), and revealed that adversarial training has manifold issues from generalization problems (Korkmaz, 2023) to black-box adversarial attacks (Korkmaz, 2024), particularly demonstrating that robust trained deep reinforcement learning policies do not have the same level of generalization skills as the vanilla trained ones (Korkmaz, 2023).

## 3 The Intrinsic Correlation of Average Negative Advantage and Directions of Instability on The Deep Neural Policy Manifold

Analyzing the functional properties and instabilities of reinforcement has long been the center of discussion (Schmidhuber, 1991; Tesauro, 1992; Singh & Dayan, 1996). In this section we will provide the theoretical analysis of the functional properties of the deep neural policy manifold. The first key quantity in our analysis is the loss $\mathcal{J}$ used to identify directions of instability on the deep neural policy manifold. Given a state-action value function $\mathcal{Q}(s, a)$ in state $s$ the argmax policy takes the action $a^*(s) = \arg\max_{a \in \mathcal{A}} \mathcal{Q}(s, a)$. The softmax policy $\pi_{\mathcal{Q}}(s, a)$ is given by the softmax function of the $\mathcal{Q}$-values for each action in state $s$ i.e. $\pi_{\mathcal{Q}}(s, a) = \frac{\exp \mathcal{Q}(s,a)}{\sum_{a' \in \mathcal{A}} \exp \mathcal{Q}(s,a')}$.

**Definition 3.1** (*Directions of Instability Loss*). The instability loss between states $s$ and $\hat{s}$ is given by the cross-entropy between the argmax policy in state $\hat{s}$ and the softmax policy in state $s$,
$$\mathcal{J}(s, \hat{s}) = -\log \pi_{\mathcal{Q}}(s, \arg\max_{a \in \mathcal{A}} \mathcal{Q}(\hat{s}, a))$$

Intuitively $\mathcal{J}(s, \hat{s})$ is a smooth measure of how far the policy in state $s$ has deviated from the softmax policy in state $\hat{s}$. Observe that since $\mathcal{Q}(s, a)$ is differentiable with respect to $s$ then so is $\mathcal{J}$. Thus, the gradient $\nabla_s \mathcal{J}(s, \hat{s})|_{s=\hat{s}}$ corresponds to the direction along which the policy most rapidly deviates from the argmax policy in state $\hat{s}$. The second quantity we study is the gap between the maximum $\mathcal{Q}$-value and the average $\mathcal{Q}$-value in each state $s$.

**Definition 3.2** (*Average Advantage Value*). The gap between the maximum and average $\mathcal{Q}$-value is called the average advantage value and represented by $\Omega(s, \hat{s})$
$$\Omega(s, \hat{s}) = \mathcal{Q}(s, \arg\max_{a \in \mathcal{A}} \mathcal{Q}(\hat{s}, a)) - \mathbb{E}_{a \sim \mathcal{A}} \mathcal{Q}(s, a)$$

Thus, the gap between the $\mathcal{Q}$-value of the optimal action in state $s$ and the average action in state $s$ is given by the evaluation of $\Omega(s, \hat{s})$ with $\hat{s} = s$ (i.e. $\Omega(s, s)$). As before, since $\mathcal{Q}(s, a)$ is differentiable with respect to $s$, then so is $\Omega(s, \hat{s})$. In this section we provide a geometric explanation on how and why there could be a correlation between average advantage value, i.e. $\Omega(s, s)$, and directions of instabilities, i.e. the norm of the gradient of the softmax cross entropy loss $\|\nabla_s \mathcal{J}(s, \hat{s})|_{s=\hat{s}}\|$.

**Proposition 3.3** (*Directions of Instability*). *Let* $\pi_{\mathcal{Q}}(s, a) = \frac{\exp \mathcal{Q}(s,a)}{\sum_{a' \in \mathcal{A}} \exp \mathcal{Q}(s,a')}$ *be the softmax policy defined by* $\mathcal{Q}$. *Let* $a^*(\hat{s}) = \arg\max_{a \in \mathcal{A}} \mathcal{Q}(\hat{s}, a)$. *Then the gradient of the cross-entropy loss* $\mathcal{J}(s, \hat{s})$ *is given by*
$$\nabla_s \mathcal{J}(s, \hat{s})|_{s=\hat{s}} = \mathbb{E}_{a \sim \pi_{\mathcal{Q}}(\hat{s},a)}[\nabla_s \mathcal{Q}(s, a)|_{s=\hat{s}}] - \nabla_s \mathcal{Q}(s, a^*(\hat{s}))|_{s=\hat{s}}$$

Hence, the proximity of the directions of the instability on the deep neural policy manifold $-\nabla_s \mathcal{J}(s, \hat{s})|_{s=\hat{s}}$ to the gradient of the advantage function $\nabla_s \Omega(s, \hat{s})|_{s=\hat{s}}$ provides the connection between $-\nabla_s \mathcal{J}(s, \hat{s})|_{s=\hat{s}}$, $\nabla_s \Omega(s, \hat{s})|_{s=\hat{s}}$ and the softmax policy $\pi_{\mathcal{Q}}(s, a)$ (see proof in A.1).

**Lemma 3.4** (*Proximity of Gradient of Average Advantage to Directions of Instability*). *Let* $\epsilon > 0$ *and suppose that* $\sum_{a \in \mathcal{A}} |\pi_{\mathcal{Q}}(\hat{s}, a) - \frac{1}{|\mathcal{A}|}| < \epsilon$. *Then*
$$\|-\nabla_s \mathcal{J}(s, \hat{s})|_{s=\hat{s}} - \nabla_s \Omega(s, \hat{s})|_{s=\hat{s}}\| < \epsilon \max_{a \in \mathcal{A}} \|\nabla_s \mathcal{Q}(s, a)|_{s=\hat{s}}\|.$$

*Proof.* By Proposition 3.3,

$$\nabla_s \mathcal{J}(s,\hat{s})|_{s=\hat{s}} = \mathbb{E}_{a \sim \pi_{\mathcal{Q}}(\hat{s},a)}[\nabla_s \mathcal{Q}(s,a)|_{s=\hat{s}}] - \nabla_s \mathcal{Q}(s, \arg\max_{a \in \mathcal{A}} \mathcal{Q}(s,a))|_{s=\hat{s}}.$$

Next, the gradient of $\Omega$ is given by

$$\nabla_s \Omega(s,\hat{s})|_{s=\hat{s}} = \nabla_s \mathcal{Q}(s, \arg\max_{a \in \mathcal{A}} \mathcal{Q}(\hat{s},a))|_{s=\hat{s}} - \frac{1}{|\mathcal{A}|}\sum_{a \in \mathcal{A}} \nabla_s \mathcal{Q}(s,a)|_{s=\hat{s}}$$

Combining the above two formulas yields

$$\|-\nabla_s \mathcal{J}(s,\hat{s})|_{s=\hat{s}} - \nabla_s \Omega(s,\hat{s})|_{s=\hat{s}}\| = \left\|\sum_{a \in \mathcal{A}}(\pi_{\mathcal{Q}}(\hat{s},a) - \frac{1}{|\mathcal{A}|})\nabla_s \mathcal{Q}(s,a)|_{s=\hat{s}}\right\|$$

$$\leq \sum_{a \in \mathcal{A}}\left\|\pi_{\mathcal{Q}}(\hat{s},a) - \frac{1}{|\mathcal{A}|}\nabla_s \mathcal{Q}(s,a)|_{s=\hat{s}}\right\| \leq \sum_{a \in \mathcal{A}}|\pi_{\mathcal{Q}}(\hat{s},a) - \frac{1}{|\mathcal{A}|}| \cdot \max_{a \in \mathcal{A}}\|\nabla_s \mathcal{Q}(s,a)|_{s=\hat{s}}\|$$

$$\leq \epsilon \|\nabla_s \mathcal{Q}(s,a)|_{s=\hat{s}}\| \qquad \square$$

Thus, Lemma 3.4 shows that a correlation between $\|\nabla_s \mathcal{J}(s,\hat{s})|_{s=\hat{s}}\|$ and $\Omega(s,s)$ is in fact a correlation between $\|\nabla_s \Omega(s,\hat{s})|_{s=\hat{s}}\|$ and $\Omega(s,s)$. Next we show that this correlation can be entirely explained by $\Omega(s,\hat{s})$ being a convex function of $s$. In particular, along every curve of steepest ascent for a strictly convex function $\Psi$, the value of $\|\nabla\Psi\|$ is increasing.

**Theorem 3.5** (*Gradient Correlation from Convexity*). *Let $\Psi : \mathbb{R}^n \to \mathbb{R}$ be strictly convex with Lipschitz continuous gradients, $s_0 \in \mathbb{R}^n$, and $\zeta > 0$. Let the curve $s : [0,\zeta] \to \mathbb{R}^n$ be the (unique) solution of $s'(\xi) = \nabla\Psi(s(\xi))$ and $s(0) = s_0$. Then for all $\xi_1, \xi_2 \in [0,\zeta]$ with $\xi_1 < \xi_2$ we have $\|\nabla\Psi(s(\xi_1))\| < \|\nabla\Psi(s(\xi_2))\|$.*

*Proof.* Note that Lipschitz continuity of $\nabla\Psi$ implies that the differential equation with boundary data $s(0) = s_0$ defining $s(\xi)$ has a unique solution. The multivariate chain rule implies that

$$\frac{d}{d\xi}\|\nabla\Psi(s(\xi))\|^2 = 2\nabla\Psi(s(\xi))^\top \left(\nabla^2\Psi(s(\xi))\right) s'(\xi). \tag{1}$$

Applying the fundamental theorem of calculus, followed by (1) yields

$$\|\nabla\Psi(s(\xi_2))\|^2 - \|\nabla\Psi(s(\xi_1))\|^2 = \int_{\xi_1}^{\xi_2} \frac{d}{d\xi}\|\nabla\Psi(s(\xi))\|^2 \, d\xi$$

$$= \int_{\xi_1}^{\xi_2} 2 \cdot \nabla\Psi(s(\xi))^\top \left(\nabla^2\Psi(s(\xi))\right) s'(\xi) d\xi = \int_{\xi_1}^{\xi_2} 2 \cdot \nabla\Psi(s(\xi))^\top \left(\nabla^2\Psi(s(\xi))\right) \nabla\Psi(s(\xi)) d\xi$$

where the final equality used the fact that $s(\xi)$ satisfies $s'(\xi) = \nabla\Psi(s(\xi))$. Next, the strict convexity of $\Psi$ implies that $\nabla^2\Psi$ is positive definite. Hence, $2 \cdot \nabla\Psi(s(\xi))^\top \left(\nabla^2\Psi(s(\xi))\right) \nabla\Psi(s(\xi)) > 0$. Thus we conclude that

$$\|\nabla\Psi(s(\xi_2))\|^2 - \|\nabla\Psi(s(\xi_1))\|^2 = \int_{\xi_1}^{\xi_2} 2 \cdot \nabla\Psi(s(\xi))^\top \left(\nabla^2\Psi(s(\xi))\right) \nabla\Psi(s(\xi)) d\xi > 0$$

which implies that $\|\nabla\Psi(s(\xi_1))\| < \|\nabla\Psi(s(\xi_2))\|$ as desired. $\square$

> *Deep reinforcement learning learns approximately-convex polices in high-dimensional MDPs.*

Theorem 3.5 shows that the observed intrinsic correlation is a direct consequence of the convexity of $\Omega(s,\hat{s})$. At first this result might seem quite surprising and counterintuitive; however, a closer examination of the training dynamics and the underlying design paradigms of deep reinforcement learning algorithms and its optimization reveals a clear underlying rationale: starting from utilization of ReLU's (Glorot et al., 2011) to network preferences (Goodfellow et al., 2012) all are specifically tuned to operate predominantly within their non-saturating regimes. Theorem 3.5 shows that along every curve of steepest ascent for $\Omega(s,s)$, the norm of the gradient of $\Omega(s,s)$ will increase. To gain further intuition on the approximate convexity of the average advantage with respect to $s$, consider two states $s, s' \in \mathcal{S}$ with the same optimal action $a^* = \arg\max_{a \in \mathcal{A}} \mathcal{Q}(s,a) = \arg\max_{a \in \mathcal{A}} \mathcal{Q}(s',a)$. For $\mathcal{Z} \in [0,1]$, let $s_{\mathcal{Z}} = (1-\mathcal{Z})s + \mathcal{Z}s'$ be the linear interpolation between $s$ and $s'$. While $s, s'$ are valid state observations of the MDP, the intermediate interpolated states $s_{\mathcal{Z}}$ are instead an unnatural, physically unrealizable convex combination of two valid states. The

$Q$-function has been trained to have a gap between the value of $a^*$ and the other actions in states both $s$ and $s'$, while the interpolated unnatural states $s_{\mathcal{Z}}$ have uncertainty on the advantage of $a^*$. Therefore, the average gap $\Omega(s_{\mathcal{Z}}, s)$ between the $Q$-value of $a^*$ and the average $Q$-value decreases from $\Omega(s, s)$ as $\mathcal{Z}$ moves away from $0$, before later increasing until it reaches the value $\Omega(s', s)$ when $\mathcal{Z} = 1$. That is, along the interpolation between $s$ and $s'$, the value of $\Omega(s_{\mathcal{Z}}, s)$ is below the straight line connecting $\Omega(s, s)$ and $\Omega(s', s)$ i.e. $\Omega((1 - \mathcal{Z})s + \mathcal{Z}s', s) \leq (1 - \mathcal{Z})\Omega(s, s) + \mathcal{Z}\Omega(s', s)$. That is, $\Omega$ is convex on the line between $s$ and $s'$. More generally, given any set of $k$ states $s_1, s_2, \ldots s_k$ in $n > k$ dimensions, the convexity of $\Omega$ holds over the simplex defined by the $k$ states, because any convex combination of the $s_i$ will on average cause the $Q$-function to be more uncertain and hence have a lower value for the average gap $\Omega$. Regarding this, Section 4.3 and Section 4.4 report results on how the intrinsic correlation of average advantage and directions of instability decreases in unnatural states from adversarial to distributional shift, and now we will prove a stronger result on convexity of $\Omega$ in high-dimensional state spaces. To do so we first need the following theorem regarding extensions of convex functions.

**Theorem 3.6** (Convex Extensions (Yan, 2014)). *Let $X$ be a compact convex subset of $\mathbb{R}^n$, and $F : \mathbb{R}^n \to \mathbb{R}$ a twice continuously differentiable function with positive definite Hessian on $X$. Then $F$ can be extended to a twice continuously differentiable function with positive definite Hessian on all of $\mathbb{R}^n$.*

Leveraging this result, we show that convexity of $\Omega$ on a simplex of states, which as described above is a result of training, implies that $\Omega$ can be extended to a convex function on the entire state space.

**Theorem 3.7** (Strict Convexity of Negative Average Advantage). *Let $S^\pi$ be a finite sequence of states by the greedy policy $\pi$ with respect to state-action value function $Q$ in an MDP $M$. Assume that the state space $\mathcal{S}$ has dimension $n \geq |S^\pi|$, and that $\Omega(s, \hat{s})$ is a twice continuously differentiable function of $s$ and strictly convex on the simplex $\mathrm{conv}(S^\pi)$. Then $\Omega$ can be extended to a strictly convex function on the entire state space.*

The proof of Theorem 3.7 is provided in the supplementary material. Intuitively, Theorem 3.7 states that if we take a trajectory of states in a high-dimensional state-space MDP, then assuming that $\Omega$ is strictly convex on the convex hull of the trajectory of states implies that $\Omega$ can be extended to a convex function on the whole state space. The approximate convexity of $\Omega$ likely arises as a byproduct of training, where actual valid state observations show a clear gap in the advantage of $a^*$ over the average action, while intermediate interpolated states are likely to lack this property.

Therefore, by Theorem 3.7 the values of $\Omega$ at the observed states in the trajectory are equal to the values of a strictly convex function which extends $\Omega$ to the entire state-space. Theorem 3.5 predict $\Omega(s, s)$ and $\|\nabla_s \Omega(s, \hat{s})|_{s=\hat{s}}\|$ to be positively correlated. These predictions further are confirmed in Section 4 in the experimental analysis. We further propose an algorithm described in Algorithm 1 to identify and diagnose the environment by leveraging this intrinsic correlation. We further investigate how we can identify adversarial disturbances, distributional shifts and even algorithmic differences via intrinsic correlations in Section 4.3 and Section 4.4.

---

**Algorithm 1** DIAL-AC: Diagnosing Deep Neural Policy Manifold via Intrinsic Correlations by Leveraging Approximate Convexity

---

**Input:** State action value function $\mathcal{Q}(s, a)$, expected value $\mu$ and standard deviation $\sigma$ of Spearman correlation from unperturbed trajectory, and tolerance $k$.
**for** $t = 1$ **to** $T$ **do**
  Set $\mathcal{G}_t = \|\nabla_s \mathcal{J}(s, s_t)|_{s=s_t}\|$ and $\Omega_t = \Omega(s_t, s_t)$
  Compute next action $a_t = \arg\max_{a \in \mathcal{A}} \mathcal{Q}(s_t, a)$.
  Observe next state $s_{t+1} \sim \mathcal{P}(s_t, a_t, \cdot)$
**end for**
Compute $\rho_{\mathrm{spearman}}$ of $\{(\Omega_t, \mathcal{G}_t)\}_{t=1}^T$.
**Return:** "Perturbation present" if $|\rho_{\mathrm{spearman}} - \mu| > k \cdot \sigma$.

---

## 4 EXPERIMENTAL ANALYSIS

The empirical analysis in this section verifies the theoretical analysis provided in Section 3. First, there is positive correlation between $\Omega(s, \hat{s})$ and $\nabla_s \mathcal{J}(s, \hat{s})$, lending support to the claim that $\Omega(s, \hat{s})$ is approximately convex as a function of $s$. Second, for trajectories of states that have been perturbed, either adversarially or by semantically meaningful perturbations to observations, the positive correlation is lower. This further confirms the theoretical arguments in Section 3, where it is predicted that the approximate convexity of $\Omega$ should be stronger at valid states that are similar to those seen in the training distribution than at perturbed states. The deep reinforcement learning

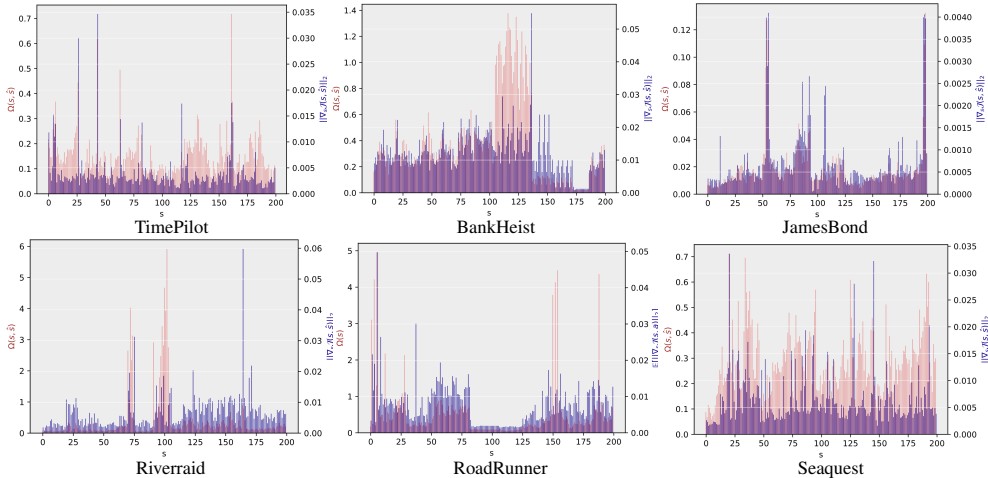

Figure 1: $\|\nabla_s \mathcal{J}(s, \hat{s})\|_2$ and $\Omega(s, \hat{s})$ values with base state observations. Results are reported with double y-axis bar graph with colors of the axis represent the same color matching the bars.

policies are trained via double $\mathcal{Q}$-learning proposed in van Hasselt (2010) and later implemented in van Hasselt et al. (2016) with dueling architecture (Wang et al., 2016) in high-dimensional MDPs. For the results reported in Section 4.1 several policies additionally are trained with the prioritized algorithm (Schaul et al., 2016). Adversarial directions are computed via the C&W formulation, ENR and Nesterov momentum. The hyperparameters for C&W are learning rate 0.01, maximum iteration 1000, initial constant 10 and for Nesterov momentum iteration is 100. Note that we used the exact hyperparameter settings for the adversarial directions to be able to provide consistent and transparent results. The policy-independent non-robust directions are computed via DCT, B&C and blur and hyperparameters were set to the exact same values as in the prior work (Korkmaz, 2024) to provide transparent and consistent comparisons.

Our empirical results will use two standard metrics to measure non-linear correlations between variables: Spearman and Kendall's tau. Spearman correlation for a sample of $n$ pairs $\{(X_i, Y_i)\}_{i=1}^n$ is computed by first calculating the ranks $R(X_i)$ and $R(Y_i)$, where $R$ assign the numerical rank $\{1, 2, \ldots n\}$ of the ranking of the $X_i$ variables from smallest to largest (and similarly for the $Y_i$ variables).

Table 1: Spearman, Kendall's $\tau$ and Pearson correlations between $\Omega(s, \hat{s})$ and $\|\nabla_s \mathcal{J}(s, \hat{s})\|_2$ with base state observations for JamesBond, Riverraid, TimePilot, Seaquest, BankHeist and RoadRunner.

| MDPs | Base Spearman | Base Kendall's $\tau$ | Base Pearson |
|---|---|---|---|
| JamesBond | 0.80887±0.01370 | 0.61751±0.01242 | 0.73822±0.01812 |
| Riverraid | 0.75758±0.00544 | 0.55706±0.00348 | 0.19795±0.01407 |
| RoadRunner | 0.66325±0.01640 | 0.46928±0.01421 | 0.40010±0.00854 |
| BankHeist | 0.81963±0.01104 | 0.62919±0.00990 | 0.53146±0.00047 |
| Seaquest | 0.63826±0.00540 | 0.45793±0.00505 | 0.56361±0.02136 |
| TimePilot | 0.94431±0.00357 | 0.81207±0.00830 | 0.86280±0.00973 |

Then the Spearman correlation is given by calculating the standard Pearson correlation of the rank variables $\rho_{R(X),R(Y)} = \frac{\text{Cov}(R(X), R(Y))}{\sigma_{R(X)} \sigma_{R(Y)}}$. Hence a perfectly montone function $Y = f(X)$ will have Spearman correlation equal to 1.0. To define Kendall's tau let $C$ denote the number of pairs of samples $(X_i, Y_i), (X_j, Y_j)$ where either $X_i > X_j$ and $Y_i > Y_j$ or $X_i < X_j$ and $Y_i < Y_j$. Let $D$ denote the number of remaining pairs, that do not satisfy the above property. Kendall's tau is given by $\tau(X, Y) = \frac{C-D}{n(n-1)/2}$. In particular, the numerator is the number of pairs where both coordinates are consistently ordered, minus the number of pairs inconsistently ordered, and the denominator is the total number of pairs. As with Spearman correlation, a perfectly monotone function $f(X, Y)$ will have $\tau(X, Y) = 1$. Table 1 reports the Spearman, Pearson and Kendall's $\tau$ results of the intrinsic correlation. These results verify the theoretical analysis provided in Section 3. Beyond the analytical probing of the deep neural policy landscape, a more intuitive understanding of neural processing can be gained by examining responses to visual illusions. This approach serves as a canonical approach in neuroscientific research (Hubel & Wiesel, 1962; Grunewald & Lankheet, 1996; Westheimer, 2008), enabling the anatomical diagnosis of processing irregularities to specific brain regions, i.e. neuro-anatomical loci, from parahippocampal cortex to cortical areas (Axelrod et al., 2017; Seymour et al., 2018). Our approach, predicated on analyzing directions of instability, draws a conceptual parallel to the neurobiological processing of visual illusionary stimuli and our

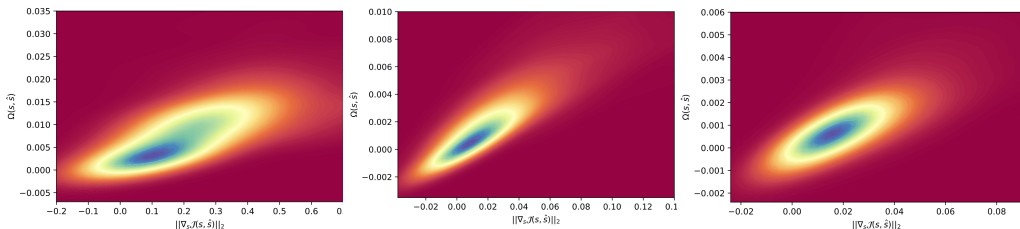

Figure 2: Heatmap results of the $\|\nabla_s \mathcal{J}(s, \hat{s})\|_2$ and $\Omega(s, \hat{s})$ values with base state observations in TimePilot, JamesBond and BankHeist.

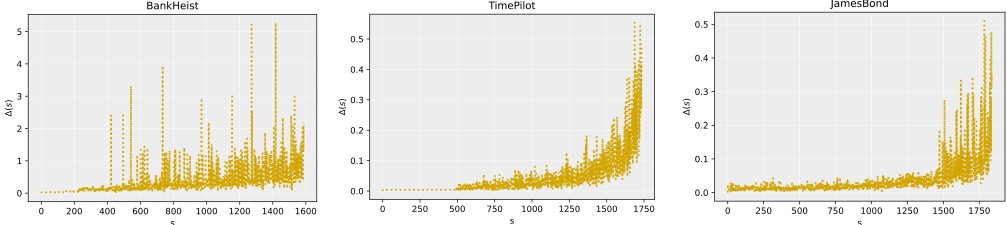

Figure 3: Positive Correlation: $\Delta(s)$ reported with base state observations in high-dimensional state representation MDPs including BankHeist, TimePilot and JamesBond.

Table 2: Architectural differences and the Spearman, Pearson and Kendall's $\tau$ correlations between $\Omega(s, \hat{s})$ and $\mathbb{E}[\|\nabla_s \mathcal{J}(s, \hat{s})\|_2]$ with base state observations.

| Environment | RoadRunner | | BankHeist | |
|---|---|---|---|---|
| Architecture | Prior | Duel | Prior | Duel |
| Spearman | 0.57537±0.048449 | **0.66325±0.016404** | 0.61924±0.011326 | **0.81963±0.0110428** |
| Kendall's $\tau$ | 0.40126±0.037366 | **0.46928±0.014211** | 0.42684±0.008932 | **0.62919±0.0099065** |
| Pearson | 0.25306±0.036695 | **0.40010±0.008546** | 0.42745±0.022079 | **0.53146±0.0004740** |

results reveal the capabilities of our method on not only the identification of the adversarial changes but further diagnosis of algorithmic and architectural differences.

## 4.1 ARCHITECTURAL AND ALGORITHMIC DIFFERENCES AND CORRELATIONS

In this section we investigate how the intrinsic correlation between directions of instability and the average advantage changes under architectural and algorithmic differences. In particular, Table 2 reports the correlation results with architectural differences between dueling and prior architecture, and the results demonstrate that the dueling architecture has higher correlation. This outcome is quite crucial because human normalized median score achieved by the dueling architecture is **172.1%** and human normalized median score obtained by prior architecture is 123.7% (Wang et al., 2016). Thus, these results further solidify that the algorithms that perform better have higher correlation between $\Omega(s, \hat{s})$ and $\|\nabla_s \mathcal{J}(s, \hat{s})\|_2$. The results reported in Table 2 demonstrate that our proposed algorithm DIAL-AC can in fact identify how algorithmic and architectural changes affect these intrinsic correlations on the deep neural policy manifold. Hence, the results in this section reveal that the approximate convexity of the advantage function is in fact higher for the algorithms that perform better, and consequently, DIAL-AC results show that the degree of approximate convexity in a function approximator represents a key property that can be targeted through architectural or algorithmic modifications to enhance the performance of reinforcement learning algorithms.

## 4.2 THE INTRINSIC CORRELATION OF ADVANTAGE AND DIRECTIONS OF INSTABILITY

Our objective is to leverage the intrinsic geometry of the deep neural policy manifold to diagnose and understand the reinforcement learning policy. Figure 1 reports $\Omega(s, \hat{s})$ and $\|\nabla_s \mathcal{J}(s, \hat{s})\|_2$ values for Riverraid, RoadRunner, Seaquest, TimePilot, JamesBond and BankHeist with base state observations. While with these results it is possible to recognize the qualitative correlation between $\Omega(s, \hat{s})$ and $\|\nabla_s \mathcal{J}(s, \hat{s})\|_2$, we further report in Table 1 the Spearman, Pearson and Kendall's $\tau$ correlations between these values. Figure 2 visualizes the kernel density estimation of points with xy-coordinates given by $\|\nabla_s \mathcal{J}(s, \hat{s})\|_2$, $\Omega(s, \hat{s})$ providing an alternative visualization of the correlation between these two quantities. To give an alternative view on the correlation, we sort the states

Table 3: Spearman and Kendall's $\tau$ correlations of $\Omega(s, \hat{s})$ and $\|\nabla_s \mathcal{J}(s, \hat{s})\|_2$ with base observations and in the existence of adversarial directions computed via C&W, ENR and Nesterov Momentum.

| Correlations | JamesBond | Riverraid | RoadRunner |
|---|---|---|---|
| Base Spearman | **0.8088733±0.013700** | **0.7575881±0.005440** | **0.663257±0.016404** |
| ElasticNet (ENR) Spearman | 0.7404537±0.007845 | 0.7312598±0.007944 | 0.360172±0.0161319 |
| Nesterov Momentum Spearman | 0.737131±0.005016 | 0.618266±0.0332335 | 0.507291±0.0103992 |
| Carlini and Wagner Spearman | 0.7774369±0.0034135 | 0.5979524±0.019333 | 0.359959±0.0443513 |
| Base Kendall's $\tau$ | **0.61751±0.012425** | **0.557067±0.003489** | **0.469280±0.014211** |
| ElasticNet (ENR) Kendall's $\tau$ | 0.5518718±0.007701 | 0.5276511±0.006757 | 0.246347±0.010676 |
| Nesterov Momentum Kendall's $\tau$ | 0.548718±0.0055495 | 0.509211±0.028019 | 0.325897±0.020279 |
| Carlini and Wagner Kendall's $\tau$ | 0.5897872±0.005087 | 0.4256483±0.014264 | 0.244640±0.030973 |
| Correlations | BankHeist | Seaquest | TimePilot |
| Base Spearman | **0.8196393±0.0110428** | **0.6382674±0.0054034** | **0.9443157±0.003573** |
| ElasticNet (ENR) Spearman | 0.6285715±0.02395142 | 0.5775322±0.0091588 | 0.9294928±0.008797 |
| Nesterov Momentum Spearman | 0.6565451±0.03250028 | 0.5542451±0.0126013 | 0.9060342±0.0140901 |
| Carlini and Wagner Spearman | 0.373751±0.0038957 | 0.4573776±0.0092046 | 0.7290117±0.006987 |
| Base Kendall's $\tau$ | **0.629199±0.0099065** | **0.4579388±0.005053** | **0.8120791±0.0083035** |
| ElasticNet (ENR) Kendall's $\tau$ | 0.4532070±0.01841255 | 0.4118854±0.0063908 | 0.7887973±0.0126924 |
| Nesterov Momentum Kendall's $\tau$ | 0.4642903±0.03478681 | 0.382152±0.0087899 | 0.7507642±0.0185963 |
| Carlini and Wagner Kendall's $\tau$ | 0.2994147±0.0393197 | 0.3155310±0.0054738 | 0.5567044±0.006622 |

$s$ in increasing by the magnitude of $\|\nabla_s \mathcal{J}(s, \hat{s})\|_2$. We then use the notation $\Delta(s)$ for $s = 1, 2, \ldots$ to indicate the value of $\Omega(s, s)$ on the 1st,2nd,... state in this sorted order. Figure 3 shows the values of $\Delta(s)$ as $s$ ranges over a trajectory of states sorted by increasing $\|\nabla_s \mathcal{J}(s, \hat{s})\|_2$, giving additional clear report of a positive correlation.

## 4.3 THE INTRINSIC CORRELATIONS UNDER ADVERSARIAL PERTURBATIONS

In this section, we investigate how the intrinsic correlation between directions of instability and the average advantage function changes under adversarial attacks. In particular, Table 3 gives the Spearman correlation and Kendall's tau results for adversarially perturbed states alongside base, i.e. unperturbed, states. These results once more verify the theoretical predictions in Section 3, where the average advantage function exhibits stronger approximate convexity at the base state observations compared to perturbed state observations. As a result, the correlation between $\Omega(s, \hat{s})$ and $\|\nabla_s \mathcal{J}(s, \hat{s})\|$ is expected to be lower for perturbed states than for base states. The clear differences in the correlation between trajectories of base states and trajectories containing perturbations allow for the detection of changes in the environment. In particular, Algorithm 1 can be used to detect when a change in the environment has occurred, and the results in Table 3 demonstrate that not only we can identify that there is an adversarial perturbation present, we can further even confirm the type of the adversarial attack by our proposed algorithm DIAL-AC. In particular, DIAL-AC precisely identifies the impact of C&W on the intrinsic correlation in contrast to the EAD attack.

## 4.4 APPROXIMATE CONVEXITY UNDER DISTRIBUTIONAL SHIFT: IMPERCEPTIBLE NATURAL NON-ROBUST PERTURBATIONS

In this section, we further investigate how correlations change under imperceptible distributional shift, i.e. natural non-robust directions. In particular, Table 4 reports Spearman and Kendall's $\tau$ correlation results under policy-independent non-robust directions computed via DCT, B&C and blur. The policy-independent non-robust directions are computed within perceptual similarity bound to ensure that these non-robust directions are as invisible as adversarial directions to the human perception. These imperceptible distributional shifts recently have been shown to break the certified robust defences in deep reinforcement learning. In particular, the perceptual similarity $\Phi_{\text{similarity}}(s, \hat{s})$ is computed via $\Phi_{\text{similarity}}(s, \hat{s}) = \sum_l \frac{1}{H_l W_l} \sum_{h,w} \|w_l \odot (\hat{y}^l_{shw} - \hat{y}^l_{\xi(s)hw})\|_2^2$ where $W_l$, $H_l$, and $C_l$ is the width, height and number of channels in the convolutional layers respectively and $\hat{y}^l_s, \hat{y}^l_{\Psi(s)} \in \mathbb{R}^{W_l \times H_l \times C_l}$ is the vector of the unit normalized activations. Figure 4 reports the values of $\Omega(s, \hat{s})$ and $\|\nabla_s \mathcal{J}(s, \hat{s})\|_2$ under different policy-independent non-robust directions. These results once more demonstrate that from adversarial attacks to distributional shift our proposed algorithm DIAL-AC can identify and diagnose the environment the reinforcement learning

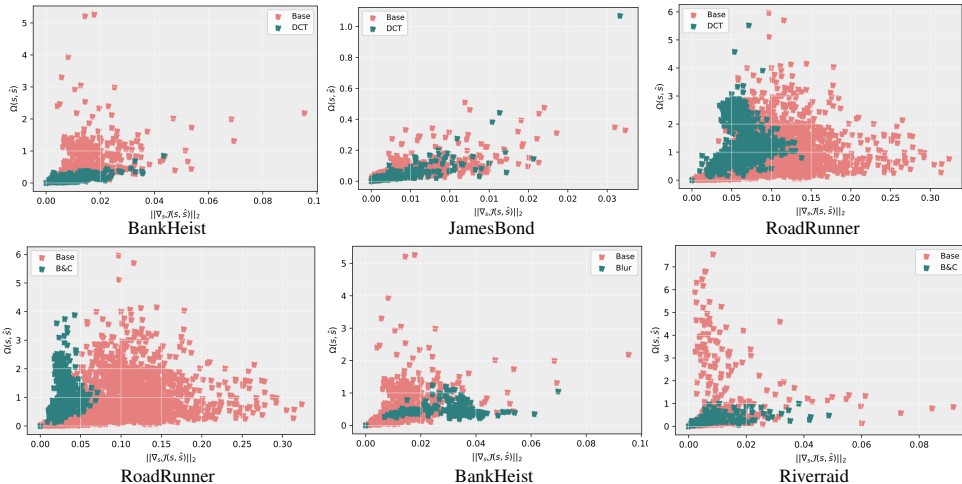

Figure 4: Scatter plots of $\Omega(s, \hat{s})$ and $\|\nabla_s \mathcal{J}(s, \hat{s})\|_2$ with base states and in the existence of imperceptible natural, i.e. policy-independent non-robust, directions computed via B&C, Blur and DCT.

Table 4: Spearman and Kendall's $\tau$ correlations between $\Omega(s, \hat{s})$ and $\|\nabla_s \mathcal{J}(s, \hat{s})\|_2$ with base state observations and in the existence of policy-independent (i.e. semantically meaningful) non-robust directions computed via DCT, B&C, Blur.

| Correlations | JamesBond | Riverraid | RoadRunner |
|---|---|---|---|
| Base Spearman | **0.808873±0.013700** | **0.757588±0.005440** | **0.66325±0.016404** |
| Blur Spearman | 0.739044±0.006811 | 0.677357±0.017423 | 0.498806±0.039418 |
| B&C Spearman | 0.711883±0.007793 | 0.670667±0.023612 | 0.384988±0.024496 |
| DCT Spearman | 0.595459±0.066479 | 0.639586±0.012965 | 0.322124±0.104422 |
| Base Kendall's $\tau$ | **0.6175157±0.012425** | **0.557067±0.003489** | **0.469280±0.014211** |
| Blur Kendall's $\tau$ | 0.545957±0.007139 | 0.507149±0.027933 | 0.340502±0.024998 |
| B&C Kendall's $\tau$ | 0.522245±0.0071492 | 0.493685±0.024185 | 0.261689±0.018721 |
| DCT Kendall's $\tau$ | 0.477147±0.0433372 | 0.454356±0.010088 | 0.218891±0.066816 |
| Correlations | BankHeist | Seaquest | TimePilot |
| Base Spearman | **0.819639±0.0110428** | **0.6382674±0.0054034** | **0.9443157±0.003573** |
| Blur Spearman | 0.2226758±0.1330715 | 0.5311349±0.0145224 | 0.9086324±0.015733 |
| B&C Spearman | 0.5663207±0.0506708 | 0.2993260±0.0580819 | 0.5897620±0.018235 |
| DCT Spearman | 0.0365417±0.1172801 | 0.4326550±0.0230028 | 0.9000135±0.008147 |
| Base Kendall's $\tau$ | **0.629199±0.0099065** | **0.4579388±0.005053** | **0.8120791±0.0083035** |
| Blur Kendall's $\tau$ | 0.1620834±0.0896341 | 0.3702977±0.0108535 | 0.7153446±0.032583 |
| B&C Kendall's $\tau$ | 0.4062786±0.0468716 | 0.2009678±0.0394226 | 0.4020167±0.004175 |
| DCT Kendall's $\tau$ | 0.0216564±0.0869163 | 0.2993766±0.0168171 | 0.7524561±0.012622 |

policy experiences, thereby allowing the policy to act according to the risks and instabilities currently present in the environment.

## 5 CONCLUSION

In this paper we focus on the underlying structure of the deep neural policy manifold and inherent functional properties of deep reinforcement learning. Our paper discovers that there is an intrinsic correlation between negative average advantage and the directions of instabilities. We provide a foundational analysis that theoretically explains the intrinsic correlation of deep neural policy and the approximate convexity by providing a mathematically rigorous investigation. We conduct extensive experiments in the Arcade Learning Environment with high-dimensional state observations. Our empirical results lay out how this intrinsic correlation changes across a wide range of adversarial attacks and natural non-robust directions, i.e. distributional shift, and further with even different training algorithms and architectural changes. Our paper further introduces an algorithm that leverages this intrinsic correlation to diagnose and understand the environment the deep neural policy is experiencing and its effects on the policy decision making. The theoretical and empirical analysis provided in our work lays out the intrinsic properties of the deep neural policy landscape that can be immediately leveraged to obtain stable and resilient policies that can make robust decisions.

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
