# OpenReview forum: "Intriguing Properties of Deep Neural Policy Manifold: Intrinsic Correlation and Deep Neural Policy Curvature"
_ICLR.cc/2026/Conference — Submitted to ICLR 2026_

### Official Review · Reviewer_8uWn · 2025-10-31

**Soundness:** 2
**Presentation:** 1
**Contribution:** 3
**Rating:** 2
**Confidence:** 2

**Summary:**

The paper investigates the correlation between the "average advantage value" and the "directions of instability" in reinforcement learning. The authors show that this correlation is weaker if the environment is perturbed through adversarial attacks or natural distribution shifts. Hence, the authors propose to monitor this property to detect such perturbations.

**Strengths:**

The topic of the paper is very relevant since robustness is an important issue in reinforcement learning. The method is evaluated on tasks from the Arcade Learning Environment, a common benchmark in reinforcement learning, demonstrating that the method is applicable to high-dimensional image-based tasks.

**Weaknesses:**

I found the paper quite hard to follow, the intuition behind the defined quantities was often unclear, and the implications of some of the theorems were not explained enough (see Questions for details). Furthermore, the setting of the experiments in section 4.4 was not clear to me.

**Questions:**

1. Definition 3.1: I understand that this is a criterion that measures how much the policy behavior changes between states $s$ and $\hat{s}$, but it is unclear to me why the criterion compares a softmax policy to an argmax policy. Why does a large discrepancy between argmax and softmax constitute a policy instability?

2. The text below Proposition 3.3 is not clear to me. Is this a consequence of Proposition 3.3? $\Omega$ does not appear in Proposition 3.3.

3. Line 177 - 178: How does that follow from Lemma 3.4?

4. Theorem 3.5: What is the relationship between $\Psi$ and $\Omega$?

5. Line 203: How does that follow from the theorem?

6. Line 205: What is meant by the "observed intrinsic correlation" here?

7. Line 215: "the intermediate interpolated states are instead an unnatural, physically unrealizable convex combination of two valid states". That should depend on the task. The paper should mention that. Does that mean that the analysis of the paper is only applicable to tasks where the convex combination of two valid states is always an invalid state?

8. Line 231: What does it mean to "extend a function"?

9. Line 365-366: "the approximate convexity of the advantage function is in fact higher for the algorithms that perform better". I don't think there is enough evidence for this claim, as the paper only compares two algorithms. The difference in the correlation could be caused by factors other than the performance.

10. Line 416: "EAD attack" is not explained in the paper.

11. The setting of these experiments is not clear to me. How are these perturbations different from those in section 4.3? In what sense are the perturbations natural and policy-independent?

12. Figure 4: How were the tasks and perturbation methods chosen here? BankHeist and RoadRunner appear twice, JamesBond and Riverraid once. DCT is present in three plots, B&C in two, and Blur in one. Why?


Comments / typos:

1. Abstract: "there is a strong correlation between the advantage function and the gradient of the loss targeting directions of instability": This sentence is very confusing for readers who have not read the main text yet. In RL, the advantage function is defined differently from the "average advantage function" of Definition 3.2. Also, "gradient of the loss targeting directions of instability" is quite hard to unpack. Perhaps there is a way to formulate this sentence in a way that relies less on the definitions in the main text?

2. Line 074: "reward function in which" --> "reward function which"

3. Line 138: I believe "softmax" should be "argmax" here (as the argmax policy is applied to state $\hat{s}$)

4. Line 155: The last ] should be a )

5. Figure 3: From the axis labels and caption alone, this plot is not clear. The meaning of $\Delta (s)$ is only explained in the text on the next page. Please consider adding more explanations to the caption or choosing more self-explanatory axis labels.

---

### Official Review · Reviewer_SCxn · 2025-10-31

**Soundness:** 2
**Presentation:** 3
**Contribution:** 2
**Rating:** 4
**Confidence:** 2

**Summary:**

This paper studies geometric properties of deep reinforcement learning (RL) policies, more specifically the relationship between the advantage function and the directions of instability in the policy’s loss landscape. It discovers a strong intrinsic correlation between \Omega(s,s) (a quantity that shows the confidence in action) and ||.\nabla_s J(s,s) || (the gradient of the instability loss measuring how the policy changes under perturbations).

**Strengths:**

The paper introduces a new and non-trivial theoretical connection between the average advantage function, and the directions of instability in the deep neural policy manifold. By proving that this correlation arises from the approximate convexity of \Omega and supporting it with formal results, the work provides a novel geometric perspective on policy behavior.

**Weaknesses:**

1. Practical Utility: Although the paper establishes a mathematical connection between convexity of \Omega(s,s) and the instability gradient, the practical utility of this correlation remains doubtful. The proposed DIAL-AC algorithm detects perturbations but does not clearly show how this information could improve training, robustness, or policy performance. Thus, the contribution is primarily diagnostic, and not something that could be used. There are no theoretical guarantees for the proposed Algorithm 1.

2. Lack of baselines: The proposed diagnostic algorithm DIAL-AC is evaluated only in isolation. It discusses correlation statistics such as Spearman, Kendall but does not compare against any baseline diagnostic or detection methods.

**Questions:**

It will be useful to explain the key result using diagrams. Also a detailed explanation on how the key result is being used algorithmically. This should also be done in the introduction. Also, how would one compute the nominal values of correlations (without attack) if non-adversarial data (interaction) is not provided?

---

### Official Review · Reviewer_8Mtw · 2025-11-01

**Soundness:** 2
**Presentation:** 3
**Contribution:** 2
**Rating:** 4
**Confidence:** 3

**Summary:**

The paper investigates the structure of deep neural policy manifolds in high-dimensional MDPs and uncovers a strong correlation between the advantage function and the gradient of the loss in directions associated with instability. Leveraging this insight, the authors propose a novel algorithm to diagnose decision volatility in deep reinforcement learning policies when exposed to environmental instabilities.

**Strengths:**

- The paper combines theoretical analysis with empirical validation, offering a geometric perspective on policy behavior in deep RL.
- The work shows that a stronger correlation between $||\nabla_s J(s,\hat{s})||$ and $\Omega(s,\hat{s})$ indicates a better performance, which could be a useful diagnostic tool.

**Weaknesses:**

- The observed positive correlation between $\|\nabla_s J(s,\hat{s})\|$ and $\Omega(s,\hat{s})$ is not particularly surprising, especially for softmax policies of the form $\pi(s,a) = \frac{\exp Q(s,a)}{\sum_{a'} \exp Q(s,a')}$, where such relationships naturally arise from the policy's sensitivity to value changes.
- Section 4.1 compares only a single pair of algorithms, which is insufficient to substantiate the broader claim that “better-performing algorithms exhibit higher correlation.” A more comprehensive comparison across more value-based methods and actor-critic methods (where the policy and value functions are not strongly connected as in softmax polies) would significantly strengthen the argument.
- There is a missing parenthesis on line 173.

**Questions:**

- In Theorem 3.5, what is $s'$ exactly? How does the theorem support the claim on line 206 that "Theorem 3.5 shows that the observed intrinsic correlation is a direct consequence of the convexity of $\Omega(s, \hat{s})$."?
- When computing the correlation between $\|\nabla_s J(s,\hat{s})\|$ and $\Omega(s,\hat{s})$, how are the state pairs $(s, s')$ (or $(s, \hat{s})$) selected? Are they sampled randomly, from trajectories, or via adversarial perturbations?
- What is the intuition behind the assertion that a stronger correlation leads to better performance? Is this correlation a cause of improved robustness, or merely a symptom of well-behaved policies? Clarifying the causal or diagnostic nature of this relationship would be valuable.

---

### Official Review · Reviewer_u8nA · 2025-11-03

**Soundness:** 1
**Presentation:** 2
**Contribution:** 2
**Rating:** 2
**Confidence:** 4

**Summary:**

This work studies the diagnosis of volatilities of DRL policy. The authors define two concepts called “Directions of Instability Loss” and “Average Advantage Value”, and show the correlation between them with formal analysis and empirical investigation (with a range of correlation analysis metrics). The experiments are conducted on several Atari environments.

**Strengths:**

- I appreciate the authors’ effort in studying the learning properties of deep neural policy network in the context of RL, which is very meaningful and significant to better understanding and addressing learning issues of DRL.

**Weaknesses:**

- The paper is overall hard to follow.
    - There are a lot of words that are not common and literally intuitive to RL community. Here are a few ones used in this paper: “directions of instability”, “natural non-robust directions to adversarial directions”. Some of them may be defined or explained in the later part of the paper, but this adds difficulties to understanding the paper smoothly.
    - The word “uncertainty" (or uncertainties) appears twice in the main body of the paper. The exact meaning remains unclear after I read the paper. For an RL researcher, naturally the word “uncertainty” relates to the epistemic uncertainty or aleatoric uncertainty in the literature of uncertainty-based RL or more general research on uncertainty quantification in machine learning.
- There are inconsistent or undefined formal notations.
    - The equation in Line 81 should be Bellman Optimality Operator ($B*$) to be distinct from Bellman (Expectation) Operator (usually denoted by $B^{\pi}$).
    - $x,y$ are not defined before being used in the equation in Line 92.
    - There are different notations of loss, i.e., $J$ Line 97 and $\mathcal{J}$ Line 137.
    - In Algorithm 1, “compute next action $a_t$”, should it be current action $a_t$? What is “unperturbed trajectory”? Are the authors referring to an adversarial or attack problem setting?
    - The definition 3.2 is closely related to the action gap [1], which should be included and discussed.
- The theoretical study is poorly motivated and the assumptions are not well justified.
    - What is the purpose of the theoretical results to prove? Why are the two conceptions defined in Section 3 important?
    - What is the reason for that Definition 3.1 uses a greedy action and a softmax policy? Since softmax policies have been defined above, it is more intuitive to define by using a distribution matrix between the two softmax policies for $s, \hat{s}$. Moreover, the metric defined in 3.1 is asymmetric.
    - In Lemma 3.4, why is $\sum_{a \ in A} |\pi_{Q}(\hat{s},a) - \frac{1}{|N|}| < \epsilon$ a reasonable assumption? What if the policy is a greedy or near greedy policy (i.e., a very common case for value-based methods and deterministic policies in AC methods)? Moreover, the greedy policy $\pi$ is also considered in Theorem 3.7.
- The experiments demonstrate that the correlation exists and the algorithms that have higher correlation between average advantage and directions of instability perform better. However, correlation does not necessarily mean causality. Whether we can leverage this critical signal to build better models, is not demonstrated in this paper and I do not think it is necessarily the case.
    - The authors claimed, “Thus, these results further solidify that the algorithms that perform better must have higher correlation". I do not think the comparison between dueling architecture and prior architecture is adequate to support the conclusion. I recommend that the authors extend their comparison by including advanced architectures, e.g., Simba [1], BroNet [2].
    - Further, “therefore the approximate convexity is an ingredient that can be targeted to make architectural or algorithmic changes to obtain higher performing reinforcement learning algorithms”, I did not find a direct empirical support to this claim (e.g., proposing a new method or improving an existing method by optimizing the “approximate convexity”.
- I suggest that the authors include more related papers that also study the geometry of function space or manifold in DRL, e.g., [4,5,6].
- There are inconsistent formats in reference, e.g., Line 523 and 530, Line 533 and 553.
- The limitation of this work is not discussed in this paper.

---
Reference

[1] Amir-massoud Farahmand. Action-Gap Phenomenon in Reinforcement Learning. NeurIPS 2011.

[2] Hojoon Lee, Dongyoon Hwang, Donghu Kim, Hyunseung Kim, Jun Jet Tai, Kaushik Subramanian, Peter R. Wurman, Jaegul Choo, Peter Stone, Takuma Seno. SimBa: Simplicity Bias for Scaling Up Parameters in Deep Reinforcement Learning. ICLR 2025.

[3] Michal Nauman, Mateusz Ostaszewski, Krzysztof Jankowski, Piotr Miłoś, Marek Cygan. Bigger, Regularized, Optimistic: scaling for compute and sample-efficient continuous control. NeurIPS 2024.

[4] Sridhar Mahadevan, Mauro Maggioni. Proto-value Functions: A Laplacian Framework for Learning Representation and Control in Markov Decision Processes. JMLR 2007.

[5] Marc G. Bellemare, Will Dabney, Robert Dadashi, Adrien Ali Taiga, Pablo Samuel Castro, Nicolas Le Roux, Dale Schuurmans, Tor Lattimore, Clare Lyle. A Geometric Perspective on  Optimal Representations for Reinforcement Learning. NeurIPS 2019.

[6] Jan Schneider, Pierre Schumacher, Simon Guist, Le Chen, Daniel Häufle, Bernhard Schölkopf, Dieter Büchler. Identifying Policy Gradient Subspaces. ICLR 2024.

**Questions:**

1. Where should the theoretical results be useful in the authors' expectation, e.g., perturbation/adversarial detection, addressing the instability or suboptimal convergence of DRL, or mitigating plasticity loss of DRL under nonstationarity? I did not find a clear theme in this paper. I feel a bit confused about the anchoring and the purpose of this work.
2. Why is $\sum_{a \ in A} |\pi_{Q}(\hat{s},a) - \frac{1}{|N|}| < \epsilon$ in Lemma 3.4 a reasonable assumption? What if the policy is a greedy policy, as also considered in Theorem 3.7.
3. Since the average advantage value (Definition 3.2) is policy independent, i.e., the second term on the right-hand side is the expectation of Q value regarding a uniform policy. How is it computed in practice (e.g., the $\Omega$ as in Algorithm 1) when the action space is continuous, e.g., MuJoCo, DMC, e.g., using distributional modeling of Q function to get the mean and how to get the argmax action? And how would it influence the theoretical results?

---

### Official Review · Reviewer_itxc · 2025-11-03

**Soundness:** 3
**Presentation:** 4
**Contribution:** 1
**Rating:** 6
**Confidence:** 2

**Summary:**

This paper identifies a strong correlation between the advantage function and loss gradients along unstable directions in deep neural policies. Leveraging this insight, the authors propose an algorithm to diagnose decision volatility in reinforcement learning policies. The method is evaluated on high-dimensional tasks in the Arcade Learning Environment, demonstrating its ability to detect sensitivity under architectural changes, distributional shifts, and worst-case perturbations.

**Strengths:**

* I command the authors for delving into better undersanding the deep RL paradigm. The discovery of the intrinsic correlation between the negative average advantage $\Omega(s, \hat{s})$ and the instability gradient norm $||\nabla_s \mathcal{J}(s, \hat{s})||$ is an interesting new insight into the functional properties of deep RL.

* Paper is well written and reasonably easy to follow for a broader audience who's not an expert in the field like myself.

* The intuitive correlation is demostrated by both theoritical proofs and extensive experiments, containing qualitative and quantitative results, establishing robustness of the findings.

**Weaknesses:**

* Limited Generalizability: The results are entirely constrained to Value-Based (DQN/Double DQN) methods in the Arcade Learning Environment. It is critical to investigate whether this geometric property extends to Policy Gradient methods (PPO, SAC, A2C) and continuous control domains (e.g., MuJoCo).

* Lack of Direct Convexity Visualization: The core theoretical justification relies on the notion of "approximate convexity," with an intuitive argument based on state interpolation ($s_{\mathcal{Z}}$). The paper would be significantly strengthened by a direct empirical visualization showing how the $\Omega(s_{\mathcal{Z}}, \hat{s})$ value behaves along a few of these interpolated lines.

* Hyperparameter Sensitivity: The practical deployment of the DIAL-AC algorithm is dependent on setting the tolerance multiplier $\kappa$. The paper lacks an analysis of $\kappa$'s sensitivity or a robust methodology for selecting a universally effective value.

* Final Contribution and Impact: Personally, I find it unclear how exactly this finding and the proposed algorithm can impact deep-RL and what it's advantages in the broader research community will be.

**Questions:**

1. Can the authors hypothesize or show preliminary results demonstrating if a similar intrinsic correlation holds for popular Policy Gradient algorithms like PPO or SAC, which operate on different loss landscapes?
2. Please provide a sensitivity analysis for the $\kappa$ hyperparameter in Algorithm 1 (DIAL-AC). How stable is the detection capability across different environments using a fixed $\kappa$ value, and what is the recommended procedure for its selection?
3. Does the intrinsic correlation consistently trend with increased network complexity (e.g., deeper or wider networks) in a standard DQN setup?
4. Does this work have any connections with the policy's state space? Do areas of the state space with high volatility (small changes in state space leads to big changes in the policy) have a correlation with the direction of instability in loss gradient?

---

### Meta-Review · Area_Chair_U3Am · 2026-01-04

**Summary:**

Here is a summary of the reviewers' concerns:
- Limited experimental justification: The experiments demonstrate that the correlation exists and the algorithms that have higher correlation between average advantage and directions of instability perform better. However, correlation does not necessarily mean causality.
- Limited theoretical justification: The theoretical study is poorly motivated and the assumptions are not well justified.
- Missing Citation and Comparison to Key Prior Work: the paper does not cite the related papers that also study the geometry of function space or manifold in DRL.
- Limited Generalizability: The word is constrained to value-based methods in the Arcade Learning Environment. It is unclear whether this geometric property extends to Policy Gradient methods and continuous control domains.
- Lack analysis of hyperparameter Sensitivity: The practical deployment of the DIAL-AC algorithm is dependent on setting the tolerance multiplier. The paper lacks an analysis of 's sensitivity or a robust methodology for selecting a universally effective value.
- Lack of baselines: The proposed diagnostic algorithm DIAL-AC is evaluated only in isolation. It discusses correlation statistics such as Spearman, Kendall but does not compare against any baseline diagnostic or detection methods.
- Unclear Contribution and Impact: it is unclear how exactly this finding and the proposed algorithm can impact deep-RL and what it's advantages in the broader research community will be.
- Not well-written: the paper is hard to follow. There are a lot of words that are not common and literally intuitive to RL community. There are inconsistent or undefined formal notations. There are inconsistent formats in reference.

**Reviewer Concerns:**

The authors did not make a rebuttal. All reviewers' concerns remain.

**Reviewer Scores:**

All reviewers are likely to keep their scores since there is no rebuttal.

---

### Decision · Program_Chairs · 2026-01-26

Reject